# Capacity and the Corresponding Heat Semigroup Characterization from Dunkl-Bounded Variation

**Xiangling Meng, Yu Liu * and Xiangyun Xie**

Department of Applied Mathematics, School of Mathematics and Physics, University of Science and Technology Beijing, Beijing 100083, China; xlmeng2021@163.com (X.M.); xiexiangyun719@163.com (X.X.)
* Correspondence: ustbmathliuyu@ustb.edu.cn

**Abstract:** In this paper, we study some important basic properties of Dunkl-bounded variation functions. In particular, we derive a way of approximating Dunkl-bounded variation functions by smooth functions and establish a version of the Gauss–Green Theorem. We also establish the Dunkl BV capacity and investigate some measure theoretic properties, moreover, we show that the Dunkl BV capacity and the Hausdorff measure of codimension one have the same null sets. Finally, we develop the characterization of a heat semigroup of the Dunkl-bounded variation space, thereby giving its relation to the functions of Dunkl-bounded variation.

**Keywords:** functions of bounded variation; Dunkl operator; Dunkl capacity; Dunkl kernel; heat kernel

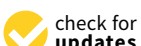



## 1. Introduction

Throughout this paper, $\Omega$ stands for an open subset of $\mathbb{R}^N$ with $N \geq 2$. We recall functions of bounded variation, which is to say functions whose weak first partial derivatives are Radon measures. Precisely, see the following definition of BV functions (cf. [1,2]).

**Definition 1.** *A function $f \in L^1(\Omega)$ is called a function of bounded variation if its total variation*

$$\|Df\|(\Omega) := \sup \left\{ \int_\Omega f \operatorname{div} \varphi \, dx : \ \varphi = (\varphi_1, \ldots, \varphi_N) \in C_c^\infty(\Omega; \mathbb{R}^N), \ |\varphi(x)| \leq 1, \ x \in \Omega \right\}$$

*is finite. The class of all such functions will be denoted by $BV(\Omega)$, where the space $BV(\Omega)$ is endowed with the norm*

$$\|f\|_{BV(\Omega)} := \|f\|_{L^1(\Omega)} + \|Df\|(\Omega).$$

Since Sobolev functions are contained within the class of BV functions of several variables, so the BV functions play an important role in some problems of variational science. For example, these function spaces have good completeness and compactness properties, consequently they are often proper settings for the applications of functional analysis, linear and nonlinear PDE theory, mathematical physics equations and fractional differential Equations (cf. [3–6]). The BV function is classically applied to the minimum area problem and the free discontinuity problem (cf. [7]). Please see [1,8–11] for more details. Another interesting and important aspect of the results is the analysis of sets of so-called finite perimeters. These sets have applications in a variety of settings due to their generality and utility. In [12], Ambrosio investigated fine properties of sets of finite perimeters in doubling metric measure spaces.

It is well known that the notion of capacity is critical in describing the null sets used to handle the pointwise behavior of Sobolev functions. The functional capacities play a significant role in every branch of mathematics, such as analysis, geometry, mathematical physics, and PDEs. Refer to [13–15] for more details. In recent years, the BV capacity has attracted the attention of many scholars. In 1989, Ziemer introduced the classical BV capacity and the related capacity inequality in [2]. In 2010, Hakkarainen and Kinnunen [16]

studied basic properties of the BV capacity and Sobolev capacity of order one in a complete metric space equipped with a doubling measure and supporting a weak Poincaré inequality. Hakkarainen and Shanmugalingam [17] further studied the relationship between the variational Sobolev 1-capacity and the BV capacity. In 2016, Xiao [18] introduced the BV-type capacity in the Gaussian space $\mathbb{G}^N$ and applied the Gaussian BV capacity to trace theory of the Gaussian BV space. In 2017, Liu [19] obtained some sharp traces and isocapacity inequalities using the BV capacity on the generalized Grushin plane $\mathbb{G}_\alpha^2$. Recently, inspired by the classical case $\alpha = 1$, Huang, Li and Liu studied the BV capacity and perimeter from the $\alpha$-Hermite bounded variation (cf. [10]). See [17,18,20,21] and the references therein for more on this topic.

Dunkl operator is found to play an increasingly important role in the study of many special functional problems with reflective symmetry. In a certain sense, the study of harmonic analysis related to the Dunkl operator is a further development of classical harmonic analysis theory and a popular research direction in the field of modern mathematics. Dunkl operator is a parameterized differential difference operator related to the finite reflection group, which operates in the Euclidean space. In recent years, these operators and their generalizations have gained considerable interest in various fields of mathematics and physics. They provide a useful tool for studying specific functions related to root systems.

Combined with the existing research results mentioned above, we know that there have been many studies of bounded variation functions in different settings due to their important roles in some problems of variational science. As we know, there are no theoretic research results in the Dunkl setting. In this paper, we will focus on the BV space, the related BV capacity and an interesting heat semigroup characterization in the context of Dunkl theory. These results may have potential applications in the theory of function spaces and PDEs in the Dunkl setting.

At first, we will present a very short introduction to Dunkl operator which is used in the following sections. The Dunkl operators were first introduced and studied by Dunkl in [22]. See [23,24] for the general theory of root systems, and see [25–28] for an overview of the rational Dunkl theory.

We shall always assume that the $N$-dimensional Euclidean space $\mathbb{R}^N$ is equipped with the standard Euclidean scalar product $\langle x, y \rangle = \sum_{j=1}^N x_j y_j$. A root system $R \subset \mathbb{R}^N \setminus \{0\}$ is a finite set if $R \cap \alpha \mathbb{R} = \{\pm \alpha\}$ and $\sigma_\alpha(R) = R$ for every $\alpha \in R$. The symbol $\sigma_\alpha$ is the reflection in the hyperplane $\langle \alpha \rangle^\perp$ orthogonal to $\alpha$, that is,

$$\sigma_\alpha(x) = x - 2\frac{\langle \alpha, x \rangle}{|\alpha|^2}\alpha,$$

where $|\alpha| = \sqrt{\langle \alpha, \alpha \rangle}$ and $x \in \mathbb{R}^N$ (cf. [24,28]).

Note that the reflection group $G = G(R)$ generated by every reflection $\sigma_\alpha$ for $\alpha \in R$ is finite, which is called the Weyl group, and is contained in the orthogonal group $O(N, \mathbb{R})$. Write a root system $R$ as the disjoint union $R = R_+ \cup (-R_+)$, where $R_+$ and $-R_+$ are separated by a hyperplane through the origin. Let us call $R_+$ a positive subsystem. Any root system can not be uniquely written as an disjoint union, but our decomposition does not affect the following definition due to the $G$-invariance of the coefficients $k$. $\mathbb{R}^N$ is divided into connected open components by the set of hyperplanes $\{\langle \alpha \rangle^\perp, \alpha \in R\}$ which is named the Weyl chambers $\mathbb{R}_e^N$. For convenience, we allow that $R$ is normalized in the sense that $\langle \alpha, \alpha \rangle = 2$ for every $\alpha \in R$.

A function $k : R \to [0, \infty)$ is called a multiplicity function on $R$, if it is invariant under the natural action of $G$ on $R$, that is, $k(\alpha) = k(g\alpha)$ for all $g \in G$ and $\alpha \in R$. The weight function $w_k$ associated with Dunkl operators on $\mathbb{R}^N$ is defined in [22,24,28] as follows:

$$w_k(x) = \prod_{\alpha \in R_+} |\langle \alpha, x \rangle|^{2k(\alpha)}. \tag{1}$$

The function $w_k$ is $G$-invariant, that is, $w_k(x) = w_k(gx)$ for all $g \in G$. Furthermore, it is also a homogeneous function of degree $2\gamma$, where

$$\gamma := \sum_{\alpha \in R_+} k(\alpha).$$

We have $k(-\alpha) = k(\alpha)$ for all $\alpha \in R$ due to the $G$-invariance of $k$. Hence, this definition does not depend on the special choice of $R_+$.

The following definitions and facts for Dunkl operator and Dunkl gradient can be seen from [28].

For $i = 1, 2, \ldots, N$, the Dunkl operator associated with $G$ on $C^1(\mathbb{R}^N)$ is defined by

$$T_i f(x) := \partial_i f(x) + \sum_{\alpha \in R_+} k(\alpha) \alpha_i \frac{f(x) - f(\sigma_\alpha x)}{\langle \alpha, x \rangle},$$

where $T_i$ denotes the directional derivative corresponding to the $i$-th standard basis vector $e_i \in \mathbb{R}^N$. The Dunkl gradient is denoted by $\nabla_k = (T_1, \ldots, T_N)$ and the Dunkl Laplacian is naturally denoted by $\Delta_k = \sum_{i=1}^N T_i^2$, more specifically,

$$\Delta_k f(x) = \Delta f(x) + 2 \sum_{\alpha \in R_+} k(\alpha) \left[ \frac{\langle \nabla f(x), \alpha \rangle}{\langle \alpha, x \rangle} - \frac{f(x) - f(\sigma_\alpha x)}{\langle \alpha, x \rangle^2} \right].$$

Notice when $k = 0$, the $T_i$ is reduced to the usual partial derivatives. At this time, $\nabla_0$ and $\Delta_0$ indicate the usual gradient and Laplacian, respectively. Denote by $L^p(\mu_k)$ the weighted space, where $d\mu_k(x) = w_k(x)\, dx$ is the Dunkl measure. The following anti-symmetry of the Dunkl operator

$$\int_{\mathbb{R}^N} T_i f(x) g(x)\, d\mu_k(x) = -\int_{\mathbb{R}^N} f(x) T_i g(x)\, d\mu_k(x) \tag{2}$$

holds for all $f, g \in C_c^1(\mathbb{R}^N)$. Moreover, we need to know an important product rule: If $f(x), g(x) \in C_c^1(\mathbb{R}^N)$, and at least one of them is $G$-invariant, then

$$T_i(f(x)g(x)) = T_i f(x) \cdot g(x) + f(x) \cdot T_i g(x). \tag{3}$$

The structure of the paper is given as follows. In Section 2, we investigate the Dunkl-bounded variation space $BV_k(\Omega)$ and obtain some basic results of BV functions belonging to $BV_k(\Omega)$, such as the structure theorem, the lower semicontinuity, approximation with smooth functions, the compactness result, and the Gauss–Green Theorem. In Section 3, we introduce the Dunkl BV capacity $\mathrm{cap}(E, BV_k(\mathbb{R}^N))$ for a set $E \subseteq \mathbb{R}^N$ and investigate the measure theoretic properties of $\mathrm{cap}(\cdot, BV_k(\mathbb{R}^N))$, we discuss the capacity of a Borel set by approximating with compact sets from inside and open sets from outside, furthermore, we study its connection to the Hausdorff measure of codimension one, which shows that the Dunkl BV capacity and the Hausdorff measure of codimension one have the same null sets. Section 4 is devoted to some results concerning the behavior of the heat semigroup for Dunkl BV functions and obtaining a heat semigroup characterization of bounded variation in the Dunkl setting. Finally, Section 5 gives a conclusion in this article.

It should be noted that, compared with the classical cases from previous works, we need to overcome some key difficulties by seeking some new methods and techniques in the proofs of our main theorems, such as, approximation with smooth functions, the Gauss–Green Theorem and the heat semigroup characterization from Dunkl-bounded variation, etc. Some key difficulties come from the difference term of the Dunkl operator. See the results in the following sections for the details.

## 2. Dunkl BV Space

Firstly, we introduce a suitable notion of functions of Dunkl bounded variation. The Dunkl divergence of a vector valued function

$$\varphi = (\varphi_1, \ldots, \varphi_N) \in C^\infty(\Omega, \mathbb{R}^N)$$

is given as follows:

$$\mathrm{div}_k \varphi(x) = \sum_{i=1}^N T_i \varphi_i(x) = \sum_{i=1}^N \left( \partial_i \varphi_i(x) + \sum_{\alpha \in R_+} k(\alpha) \alpha_i \frac{\varphi_i(x) - \varphi_i(\sigma_\alpha x)}{\langle \alpha, x \rangle} \right).$$

The Dunkl variation of $f \in L^1(\Omega, \mu_k)$ is defined by

$$\|\nabla_k f\|(\Omega) = \sup_{\varphi \in \mathcal{F}(\Omega, \mathbb{R}^N)} \left\{ \int_\Omega f(x) \mathrm{div}_k \varphi(x) \, d\mu_k(x) \right\},$$

where

$$\mathcal{F}(\Omega, \mathbb{R}^N) := \left\{ \varphi = (\varphi_1, \ldots, \varphi_N) \in C_c^\infty(\Omega, \mathbb{R}^N), \|\varphi\|_{L^\infty} \leq 1 \right\},$$

and

$$\|\varphi\|_{L^\infty} = \sup_{x \in \Omega} (|\varphi_1(x)|^2 + \cdots + |\varphi_N(x)|^2)^{\frac{1}{2}}.$$

A $L^1(\mu_k)$ function is said to have the Dunkl-bounded variation on $\Omega$ if

$$\|\nabla_k f\|(\Omega) < \infty,$$

and the set of all of these functions is denoted as $BV_k(\Omega)$, which is a Banach space. We will prove it in detail later in Lemma 5. If we choose $k = 0$, then it is the classical BV space.

A function $f \in L^1_{\mathrm{loc}}(\Omega, \mu_k)$ has locally Dunkl-bounded variation in $\Omega$ if for each open set $U \subset\subset \Omega$,

$$\|\nabla_k f\|(U) := \sup \left\{ \int_U f(x) \mathrm{div}_k \varphi(x) \, d\mu_k(x) | \varphi \in C_c^\infty(U, \mathbb{R}^N), \|\varphi\|_{L^\infty} \leq 1 \right\} < \infty.$$

We use $BV_{k,\mathrm{loc}}(\Omega)$ to denote the space of such functions.

Let $\Omega \subset \mathbb{R}^N$ be a bounded open set and $E \subset \Omega$ be a Borel set. We can define

$$\|\nabla_k f\|(E) = \inf\{\|\nabla_k f\|(U) : E \subset U, U \subset \Omega \text{ open}\}.$$

In fact, $\|\nabla_k f\|(\cdot)$ is a Radon measure in $\Omega$. Let us prove the general structure theorem.

**Lemma 1.** *(Structure Theorem for $BV_k$ functions). Let $f \in BV_k(\Omega)$. Then there exists a Radon measure $\mu$ on $\Omega$ such that*

$$\int_\Omega f(x) \, \mathrm{div}_k \varphi(x) \, d\mu_k(x) = - \int_\Omega \varphi(x) \cdot d\mu(x)$$

*for every $\varphi \in C_c^\infty(\Omega, \mathbb{R}^N)$ and*

$$\|\nabla_k f\|(\Omega) = |\mu|(\Omega),$$

*where $|\mu|$ is the total variation of the measure $\mu$.*

**Proof.** It is easy to see that

$$\left| \int_\Omega f(x) \, \mathrm{div}_k \varphi(x) \, d\mu_k(x) \right| \leq \|\nabla_k f\|(\Omega) \|\varphi\|_{L^\infty(\Omega)} \quad \forall \varphi \in C_c^\infty(\Omega, \mathbb{R}^N).$$

Denote by the functional $\Phi$ with

$$\Phi : C_c^\infty(\Omega, \mathbb{R}^N) \to \mathbb{R},$$

where

$$\langle \Phi, \varphi \rangle := \int_\Omega f(x) \, \text{div}_k \varphi(x) \, d\mu_k(x).$$

Then using the Hahn-Banach theorem, we know that there exists a linear and continuous extension $L$ of $\Phi$ to the normed space $(C_c(\Omega, \mathbb{R}^N), \|\cdot\|_{L^\infty(\Omega)})$ such that

$$\|L\| = \|\Phi\| = \|\nabla_k f\|(\Omega).$$

By the Riesz representation theorem (cf. Corollary 1.55 in [7]), there exists a unique $\mathbb{R}^N$-valued finite Radon measure $\mu$ such that

$$L(\varphi) = \int_\Omega \varphi(x) \cdot d\mu(x) \quad \forall \varphi \in C_c(\Omega, \mathbb{R}^N),$$

and $|\mu|(\Omega) = \|L\|$. Thus, we have $|\mu|(\Omega) = \|\nabla_k f\|(\Omega)$. This completes the proof. $\quad \square$

As we know, $W_{\text{loc}}^{1,1}(\Omega) \subset BV_{\text{loc}}(\Omega)$ implies that each Sobolev function has bounded variation. We similarly obtain the relation in the Dunkl setting. The Dunkl Sobolev space $W_k^{1,p}(\Omega)$ is the space consisting of all functions $f \in L^p(\mu_k)$, $1 \le p \le \infty$ and $\nabla_k f \in L^p(\mu_k)$ in a weak sense. The norm of $f \in W_k^{1,p}(\Omega)$ is defined as

$$\|f\|_{W_k^{1,p}(\Omega)} := \left( \int_\Omega \left( |f(x)|^p + |\nabla_k f(x)|^p \right) d\mu_k(x) \right)^{\frac{1}{p}}.$$

**Lemma 2.** *(Local inclusion of Sobolev functions). If $\Omega \subset \mathbb{R}^N$ is an open set, then*

$$W_{k,\text{loc}}^{1,1}(\Omega) \subset BV_{k,\text{loc}}(\Omega).$$

**Proof.** Suppose $f \in W_{k,\text{loc}}^{1,1}(\Omega)$, $U \subset\subset \Omega$ open and let $\varphi \in C_c^\infty(U, \mathbb{R}^N)$ with $\|\varphi\|_{L^\infty(U)} \le 1$. Then using (2), we have

$$\int_U f(x) \text{div}_k \varphi(x) \, d\mu_k(x) = -\int_U \nabla_k f(x) \cdot \varphi(x) \, d\mu_k(x) \le \int_U |\nabla_k f| \, d\mu_k(x) < \infty.$$

Taking the supremum over $\varphi$, we can derive the proof of the lemma. $\quad \square$

Next we show that for every $W_k^{1,1}(\Omega)$ function, the Dunkl-bounded variation $\|\nabla_k f\|(\Omega)$ boils down to the usual local Dunkl norm.

**Lemma 3.** *($BV_k$ norm on $W_k^{1,1}$). If $f \in W_k^{1,1}(\Omega)$, then*

$$\|\nabla_k f\|(\Omega) = \int_\Omega |\nabla_k f(x)| \, d\mu_k(x).$$

*Furthermore, if $f \in BV_k(\Omega) \cap C^\infty(\Omega)$, then $f \in W_k^{1,1}(\Omega)$.*

**Proof.** If $f \in W_k^{1,1}(\Omega)$, we know that $\nabla_k f(x) \in L^1(\Omega, \mu_k)$. For each $\varphi \in C_c^\infty(\Omega, \mathbb{R}^N)$ with $\|\varphi\|_{L^\infty(\Omega)} \le 1$, we get

$$\left| \int_\Omega f(x) \text{div}_k \varphi(x) \, d\mu_k(x) \right| = \left| \int_\Omega \nabla_k f(x) \cdot \varphi(x) \, d\mu_k(x) \right| \le \int_\Omega |\nabla_k f(x)| \, d\mu_k(x).$$

By taking the supremum over $\varphi$, we prove $f \in BV_k(\Omega, \mathbb{R}^N)$ and

$$\|\nabla_k f\|(\Omega) \leq \int_\Omega |\nabla_k f(x)| \, d\mu_k(x).$$

Then we prove the reverse one is valid. Now define $g \in L^\infty(\Omega, \mathbb{R}^N)$ by setting

$$g(x) := \begin{cases} \dfrac{\nabla_k f(x)}{|\nabla_k f(x)|}, & \text{if } x \in \Omega \text{ and } |\nabla_k f(x)| \neq 0, \\ 0, & \text{otherwise.} \end{cases}$$

It is easy to see that $\|g\|_{L^\infty} \leq 1$. By a standard approximation result, there exists a sequence $\{\varphi_n\}_{n \in \mathbb{N}} \subset C_c^\infty(\Omega, \mathbb{R}^N)$ such that $\varphi_n \to g$ pointwise as $n \to \infty$, with $\|\varphi_n\|_{L^\infty(\Omega)} \leq 1$ for all $n \in \mathbb{N}$. Considering the definition of $\|\nabla_k f\|(\Omega)$, after integration by parts, and then for each $n \geq 1$, we have

$$\|\nabla_k f\|(\Omega) \geq -\sum_{i=1}^N \int_\Omega T_i f(x) \cdot \varphi_n^{(i)}(x) \, d\mu_k(x) = -\int_\Omega \nabla_k f(x) \cdot \varphi_n(x) \, d\mu_k(x),$$

where $\varphi_n(x) = (\varphi_n^{(1)}(x), \ldots, \varphi_n^{(N)}(x))$. By the dominated convergence theorem and the definition of $g$, when $n \to \infty$ we obtain

$$\|\nabla_k f\|(\Omega) \geq \int_\Omega |\nabla_k f(x)| \, d\mu_k(x),$$

so we complete the proof of the first statement.

If $f \in BV_k(\Omega) \cap C^\infty(\Omega)$, fix a compact set $O \subset \Omega$ with nonempty interior and define $g_O := g\chi_{\text{int}(O)}$. Similarly to the previous arguments, we can find a sequence $\{\varphi_n\}_{n \in \mathbb{N}} \subset C_c^\infty(\text{int}(O), \mathbb{R}^N)$ so that $\varphi_n \to g_O$ pointwise with $\|\varphi_n\|_{L^\infty(\text{int}(O))} \leq 1$ for all $n \in \mathbb{N}$. Thus, we get

$$\begin{aligned}
\|\nabla_k f\|(\Omega) &\geq \int_\Omega f(x) \, \text{div}_k \varphi_n(x) \, d\mu_k(x) \\
&= \int_O f(x) \, \text{div}_k \varphi_n(x) \, d\mu_k(x) \\
&= -\sum_{i=1}^N \int_O T_i f(x) \cdot \varphi_n^{(i)}(x) \, d\mu_k(x).
\end{aligned}$$

Since $f \in C^\infty(\Omega)$, then $\nabla_k f \in L^1(O, \mu_k)$. Consequently,

$$\|\nabla_k f\|(\Omega) \geq \int_O |\nabla_k f| \, d\mu_k(x),$$

where we have used the dominated convergence theorem. Finally, the proof is completed using an exhaustive sequence of compacts via monotone convergence. $\square$

**Lemma 4.** *(Lower semicontinuity of Dunkl variation).* Suppose $f_n \in BV_k(\Omega), n = 1, 2, \ldots,$ and $f_n \to f$ in $L^1_{\text{loc}}(\Omega, \mu_k)$. Then

$$\liminf_{n \to \infty} \|\nabla_k f_n\|(\Omega) \geq \|\nabla_k f\|(\Omega).$$

**Proof.** Fix $\varphi \in C_c^\infty(\Omega, \mathbb{R}^N)$ with $\|\varphi\|_{L^\infty(\Omega)} \leq 1$. Firstly, we use the definition of $\|\nabla_k f_n\|(\Omega)$ to get

$$\|\nabla_k f_n\|(\Omega) \geq \int_\Omega f_n(x) \text{div}_k \varphi(x) \, d\mu_k(x).$$

Since $\{f_n\}_{n\in\mathbb{N}}$ converges to $f$ in $L^1_{\text{loc}}(\Omega, \mu_k)$, then via the dominated convergence theorem, we can obtain

$$\liminf_{n\to\infty} \|\nabla_k f_n\|(\Omega) \geq \int_\Omega f(x) \operatorname{div}_k \varphi(x) \, d\mu_k(x).$$

Then, according the arbitrariness of such functions $\varphi$ and the definition of $\|\nabla_k f\|(\Omega)$, we can get the conclusion.  □

**Lemma 5.** *The space $\left(BV_k(\Omega), \|\cdot\|_{BV_k(\Omega)}\right)$ is a Banach space.*

**Proof.** It is easy to check that $\|\cdot\|_{BV_k(\Omega)}$ is a norm and we omit the details. In what follows, we prove the completeness of $BV_k(\Omega)$. Let $\{f_n\}_{n\in\mathbb{N}} \subset BV_k(\Omega)$ be a Cauchy sequence, namely, for every $\varepsilon > 0$ there exists $n_0 \in \mathbb{N}$ such that $\forall n, m \geq n_0$, we have

$$\|\nabla_k(f_m - f_n)\|(\Omega) < \varepsilon.$$

Especially, $\{f_n\}_{n\in\mathbb{N}}$ is a Cauchy sequence in the Banach space $(L^1(\Omega, \mu_k), \|\cdot\|_{L^1(\Omega,\mu_k)})$, which implies that there exists $f \in L^1(\Omega, \mu_k)$ with $\|f_n - f\|_{L^1(\Omega,\mu_k)} \to 0$ as $n \to \infty$. Hence, via Lemma 4, we have

$$\|\nabla_k(f - f_m)\|(\Omega) \leq \liminf_n \|\nabla_k(f_m - f_n)\|(\Omega) \leq \varepsilon \quad \forall m \geq n_0,$$

which implies that $\|\nabla_k(f_m - f)\|(\Omega) \to 0$ as $m \to \infty$. This completes the proof.  □

Next we will consider the approximation by smooth functions for functions of Dunkl-bounded variation.

**Lemma 6.** *(Approximation with smooth functions). Assume $f \in BV_k(\Omega)$, there exists a sequence of functions $\{f_n\}_{n=1}^\infty \subset BV_k(\Omega) \cap C_c^\infty(\Omega)$ such that*
(i)   *$f_n \to f$ in $L^1(\Omega, \mu_k)$;*
(ii)   *$\int_\Omega |\nabla_k f_n(x)| \, d\mu_k(x) \to \|\nabla_k f\|(\Omega)$ as $n \to \infty$.*

**Proof.** We adapt the method of the proof in Theorem 5.3 in [1] to prove this theorem. Firstly, via the lower semicontinuity property of $BV_k$ functions established in Lemma 4, we only need to prove that, for $f \in BV_k(\Omega)$ and every $\varepsilon > 0$, there exists a function $f_\varepsilon \in C^\infty(\Omega)$ such that

$$\int_\Omega |f(x) - f_\varepsilon(x)| \, d\mu_k(x) < \varepsilon, \quad \text{and} \quad \|\nabla_k f_\varepsilon\|(\Omega) < \|\nabla_k f\|(\Omega) + \varepsilon.$$

Define a sequence of open sets, for $m \in \mathbb{N}$,

$$\Omega_j := \left\{x \in \Omega \mid \operatorname{dist}(x, \partial\Omega) > \frac{1}{m+j}\right\} \cap B(0, t+m),$$

where $j \in \mathbb{N}$ and $B(0, t+m)$ is an open ball of center 0 and radius $t+m$, $\operatorname{dist}(x, \partial\Omega)$ denotes the Euclidean distance from $x$ to $\partial\Omega$. Since $\|\nabla_k f\|(\cdot)$ is a Radon measure, for $\varepsilon > 0$, we can choose $m \in \mathbb{N}$ so large that

$$\|\nabla_k f\|(\Omega \setminus \Omega_0) < \varepsilon. \tag{4}$$

Note that the sequence of open domains $\{\Omega_j\}$ satisfies the following ways:

$$\Omega_j \subset \Omega_{j+1} \subset \Omega \quad \text{for any } j \in \mathbb{N}, \quad \text{and} \quad \bigcup_{j=0}^\infty \Omega_j = \Omega.$$

We consider another sequence of open sets

$$U_0 := \Omega_0, \quad U_j := \Omega_{j+1} \setminus \overline{\Omega}_{j-1} \quad \text{for } j \geq 1.$$

By standard results, there exists a partition of unity related to the covering $\{U_j\}_{j \in \mathbb{N}}$, which shows that there exists $\{\zeta_j\}_{j \in \mathbb{N}} \in C_c^\infty(U_j)$ such that $0 \leq \zeta_j \leq 1$ for every $j \geq 0$ and $\sum_{j=0}^\infty \zeta_j = 1$ on $\Omega$. Namely, we have the following fact:

$$\sum_{j=0}^\infty \nabla_k \zeta_j(x) = \sum_{j=0}^\infty \nabla \zeta_j(x) + \sum_{i=1}^N \sum_{\alpha \in R_+} k(\alpha) \alpha_i \frac{\sum_{j=0}^\infty \zeta_j(x) - \sum_{j=0}^\infty \zeta_j(\sigma_\alpha x)}{\langle \alpha, x \rangle} = 0 \quad \text{on } \Omega. \quad (5)$$

Let $\eta \in C_c^\infty(\mathbb{R}^N)$ be a radial nonnegative function with $\int_{\mathbb{R}^N} \eta(x) \, d\mu_k(x) = 1$ and $\mathrm{supp}(\eta) \subset B(0,1)$. Given $\varepsilon > 0$ and $f \in L^1(\Omega, \mu_k)$, extended to zero out of $\Omega$, we define the following regularization:

$$f_\varepsilon(x) = \frac{1}{\varepsilon^{N+2\gamma}} \int_{B(x,\varepsilon)} \eta(\frac{x-y}{\varepsilon}) f(y) \, d\mu_k(y).$$

We can easily conclude that for each $j \geq 0$ there exists $0 < \varepsilon_j < \varepsilon$ such that

$$\begin{cases} \mathrm{supp}((\zeta_j f)_{\varepsilon_j}(x)) \subseteq U_j; \\ \int_\Omega | (\zeta_j f)_{\varepsilon_j}(x) - \zeta_j(x) f(x) | \, d\mu_k(x) < \varepsilon \, 2^{-(j+1)}; \\ \int_\Omega | (f \nabla_k \zeta_j)_{\varepsilon_j}(x) - f(x) \nabla_k \zeta_j(x) | \, d\mu_k(x) < \varepsilon \, 2^{-(j+1)}. \end{cases} \quad (6)$$

Now define $f_\varepsilon := \sum_{j=0}^\infty (f \zeta_j)_{\varepsilon_j}$. In some neighborhood of each point $x \in \Omega$ and the sum is locally finite, then we get that $f_\varepsilon \in C^\infty(\Omega)$ and $f = \sum_{j=0}^\infty f \zeta_j$. (6) implies

$$\|f_\varepsilon - f\|_{L^1(\Omega, \mu_k)} \leq \sum_{j=0}^\infty \int_\Omega |\eta_{\varepsilon_j} * (f \zeta_j)(x) - f(x) \zeta_j(x)| \, d\mu_k(x) < \varepsilon.$$

Consequently,

$$f_\varepsilon \to f \quad \text{in} \quad L^1(\Omega, \mu_k) \quad \text{as} \quad \varepsilon \to 0.$$

Moreover, the lower semicontinuity of Dunkl variation in Lemma 4 implies that

$$\|\nabla_k f\|(\Omega) \leq \liminf_{\varepsilon \to 0} \|\nabla_k f_\varepsilon\|(\Omega). \quad (7)$$

Suppose $\varphi \in C_c^\infty(\Omega, \mathbb{R}^N)$, $\|\varphi\|_{L^\infty} \leq 1$. We start a direct computation and get

$$\int_\Omega f_\varepsilon(x) \mathrm{div}_k \varphi(x) \, d\mu_k(x) = \sum_{j=0}^\infty \int_\Omega ((f\zeta_j) * \eta_{\varepsilon_j})(x) \, \mathrm{div}_k \varphi(x) \, d\mu_k(x)$$

$$= \sum_{j=0}^\infty \int_\Omega \int_\Omega \frac{1}{\varepsilon_j^{N+2\gamma}} \eta(\frac{x-y}{\varepsilon_j}) f(y) \, \zeta_j(y) \, \mathrm{div}_k \varphi(x) \, d\mu_k(y) d\mu_k(x)$$

$$= \sum_{j=0}^\infty \int_\Omega f(x) \, \mathrm{div}_k(\zeta_j(\eta_{\varepsilon_j} * \varphi))(x) \, d\mu_k(x)$$

$$- \sum_{j=0}^\infty \int_\Omega f(x) \, \nabla_k \zeta_j \cdot (\eta_{\varepsilon_j} * \varphi)(x) \, d\mu_k(x)$$

$$= \sum_{j=0}^\infty \int_\Omega f(x) \, \mathrm{div}_k(\zeta_j(\eta_{\varepsilon_j} * \varphi))(x) \, d\mu_k(x)$$

$$- \sum_{j=0}^\infty \int_\Omega \varphi(x) \cdot (\eta_{\varepsilon_j} * (f \nabla_k \zeta_j) - f \nabla_k \zeta_j)(x) \, d\mu_k(x)$$

$$:= I + II,$$

where we have used (5) in the fourth step and the third equal sign above holds due to the fact that if $f \in C_c^\infty(\Omega)$, $g = (g_1, \ldots, g_N) \in C_c^\infty(\Omega, \mathbb{R}^N)$, then

$$\int_\Omega \operatorname{div}_k(f(x)g(x)) \, d\mu_k(x) = \int_\Omega f(x) \operatorname{div}_k g(x) \, d\mu_k(x) + \int_\Omega \nabla_k f(x) \cdot g(x) \, d\mu_k(x).$$

In fact, for $i = 1, \ldots, N$,

$$\int_\Omega f(x) \, T_i g_i(x) + g_i(x) \, T_i f(x) - T_i(f(x)g_i(x)) \, d\mu_k(x)$$

$$= \int_\Omega \sum_{\alpha \in R_+} k(\alpha)\alpha_i \frac{f(x)g_i(x) + f(\sigma_\alpha x)g_i(\sigma_\alpha x) - f(x)g_i(\sigma_\alpha x) - g_i(x)f(\sigma_\alpha x)}{\langle \alpha, x \rangle} \, d\mu_k(x)$$

$$= \int_\Omega \sum_{\alpha \in R_+} k(\alpha)\alpha_i \frac{f(x)g_i(x)}{\langle \alpha, x \rangle} \, d\mu_k(x) - \int_\Omega \sum_{\alpha \in R_+} k(\alpha)\alpha_i \frac{f(\sigma_\alpha x)g_i(\sigma_\alpha x)}{\langle \alpha, \sigma_\alpha x \rangle} \, d\mu_k(x)$$

$$- \int_\Omega \sum_{\alpha \in R_+} k(\alpha)\alpha_i \frac{f(x)g_i(\sigma_\alpha x)}{\langle \alpha, x \rangle} \, d\mu_k(x) + \int_\Omega \sum_{\alpha \in R_+} k(\alpha)\alpha_i \frac{f(\sigma_\alpha x)g_i(x)}{\langle \alpha, \sigma_\alpha x \rangle} \, d\mu_k(x)$$

$$= 0,$$

where we have used the fact $\langle \alpha, x \rangle = -\langle \alpha, \sigma_\alpha x \rangle$ and the $G$-invariance: $w_k(x) = w_k(\sigma_\alpha x)$. We note that $\|\zeta_j(\eta_{\varepsilon_j} * \varphi)\|_{L^\infty} \leq 1$ $(j = 1, \ldots)$. Recalling that by construction every point $x \in \Omega$ belongs to at most three of the sets $\{U_j\}_{j=1}^\infty$, then we have

$$|I| = \left| \int_\Omega f(x) \operatorname{div}_k\big(\zeta_1(\eta_{\varepsilon_1} * \varphi)\big)(x) \, d\mu_k(x) + \sum_{j=2}^\infty \int_\Omega f(x) \operatorname{div}_k\big(\zeta_j \eta_{\varepsilon_j} * \varphi\big)(x) \, d\mu_k(x) \right|$$

$$\leq \|\nabla_k f\|(\Omega) + \sum_{j=2}^\infty \|\nabla_k f\|(U_j)$$

$$\leq \|\nabla_k f\|(\Omega) + 3\|\nabla_k f\|(\Omega \setminus \Omega_0)$$

$$\leq \|\nabla_k f\|(\Omega) + 3\varepsilon,$$

where the last inequality is given by using (4). On the other hand, we use (6) to obtain

$$|II| < \varepsilon.$$

Then

$$\int_\Omega f_\varepsilon(x) \operatorname{div}_k \varphi(x) \, d\mu_k(x) \leq \|\nabla_k f\|(\Omega) + 4\varepsilon,$$

which implies that

$$\|\nabla_k f_\varepsilon\|(\Omega) \leq \|\nabla_k f\|(\Omega) + 4\varepsilon.$$

Now we use the above estimate and (7) to complete the proof. □

**Lemma 7.** *(max–min property of the Dunkl variation). If $f, g \in L^1(\Omega, \mu_k)$. Then*

$$\|\nabla_k \max\{f, g\}\|(\Omega) + \|\nabla_k \min\{f, g\}\|(\Omega) \leq \|\nabla_k f\|(\Omega) + \|\nabla_k g\|(\Omega).$$

**Proof.** Firstly, we can suppose that

$$\|\nabla_k f\|(\Omega) + \|\nabla_k g\|(\Omega) < \infty,$$

otherwise, the conclusion is clearly true when $\|\nabla_k f\|(\Omega) + \|\nabla_k g\|(\Omega) = \infty$. Choose functions

$$f_m, g_m \in C_c^\infty(\Omega) \cap BV_k(\Omega), \quad m = 1, 2, \ldots,$$

such that

$$\begin{cases} f_m \to f, g_m \to g \quad \text{in} \quad L^1(\Omega, \mu_k); \\ \int_\Omega |\nabla_k f_m(x)| \, d\mu_k(x) \to \|\nabla_k f\|(\Omega); \\ \int_\Omega |\nabla_k g_m(x)| \, d\mu_k(x) \to \|\nabla_k g\|(\Omega). \end{cases}$$

As

$$\max\{f_m, g_m\} \to \max\{f, g\} \quad \text{and} \quad \min\{f_m, g_m\} \to \min\{f, g\} \quad \text{in} \quad L^1(\Omega, \mu_k),$$

it follows that

$$\begin{aligned} \|\nabla_k \max\{f, g\}\|(\Omega) &+ \|\nabla_k \min\{f, g\}\|(\Omega) \\ &\leq \liminf_{m \to \infty} \int_\Omega |\nabla_k \max\{f_m, g_m\}| \, d\mu_k(x) \\ &\quad + \liminf_{m \to \infty} \int_\Omega |\nabla_k \min\{f_m, g_m\}| \, d\mu_k(x) \\ &\leq \liminf_{m \to \infty} \left( \int_\Omega |\nabla_k \max\{f_m, g_m\}| \, d\mu_k(x) \right. \\ &\quad \left. + \int_\Omega |\nabla_k \min\{f_m, g_m\}| \, d\mu_k(x) \right) \\ &\leq \lim_{m \to \infty} \int_\Omega |\nabla_k f_m(x)| \, d\mu_k(x) + \lim_{m \to \infty} \int_\Omega |\nabla_k g_m(x)| \, d\mu_k(x) \\ &= \|\nabla_k f\|(\Omega) + \|\nabla_k g\|(\Omega). \end{aligned}$$

□

**Lemma 8.** *(Compactness for $BV_k(\Omega)$). Suppose $\Omega \subset \mathbb{R}^N$ is an open and bounded domain with Lipschitz boundary. Let $\{f_j\}_{j=1}^\infty$ be a sequence in $BV_k(\Omega)$ satisfying*

$$\sup_j \left( \|f_j\|_{L^1(\Omega, \mu_k)} + \|\nabla_k f_j\|(\Omega) \right) < \infty.$$

*Then there exists a subsequence $\{f_{j_m}\}_{m=1}^\infty$ and a function $f \in BV_k(\Omega)$ such that $f_{j_m} \to f$ in $L^1(\Omega, \mu_k)$ when $m \to \infty$.*

**Proof.** According to approximation with smooth functions, for $j = 1, 2, \ldots$, there is a sequence $g_j \in BV_k(\Omega) \cap C^\infty(\Omega)$ such that

$$\begin{cases} \int_\Omega |f_j(x) - g_j(x)| \, d\mu_k(x) < \frac{1}{j}, \\ \sup_j \int_\Omega |\nabla_k g_j(x)| \, d\mu_k(x) < \infty. \end{cases} \tag{8}$$

In particular,

$$\int_\Omega |g_j(x)| \, d\mu_k(x) \leq \int_\Omega |f_j(x) - g_j(x)| \, d\mu_k(x) + \int_\Omega |f_j(x)| \, d\mu_k(x) \leq \sup_j \|f_j\|_{L^1(\Omega, \mu_k)} + 1.$$

Now, we get that $g_j \in W_k^{1,1}(\Omega)$ and

$$\int_\Omega |\nabla_k g_j(x)| \, d\mu_k(x) = \|\nabla_k g_j\|(\Omega).$$

Therefore, $\{g_j\}_{j=1}^{\infty}$ is a bounded sequence in $W_k^{1,1}(\Omega)$. Since $\Omega$ has smooth boundary, it follows from Rellich's compact embedding theorem that there exists $f \in L^1(\Omega, \mu_k)$ and a subsequence $\{g_{j_m}\}_{m=1}^{\infty}$ such that $g_{j_m} \to f$ in $L^1(\Omega, \mu_k)$. Then from (8), we know $f_{j_m} \to f$ in $L^1(\Omega, \mu_k)$. By lower semicontinuity of Dunkl variation, we obtain

$$\|\nabla_k f\|(\Omega) \leq \liminf_{j_m}\|\nabla_k f_{j_m}\|(\Omega) \leq \sup_j\|\nabla_k f_j\|(\Omega) < \infty,$$

which shows that $f \in BV_k(\Omega)$ and this completes the proof. $\quad\square$

Naturally, we can give the perimeter of a set in the Dunkl setting.

**Definition 2.** *The Dunkl perimeter of $E \subseteq \Omega$ is defined as:*

$$P_k(E, \Omega) = \|\nabla_k \chi_E\|(\Omega) = \sup_{\varphi \in \mathcal{F}(\Omega, \mathbb{R}^N)} \left\{ \int_E \mathrm{div}_k \varphi(x)\, d\mu_k(x) \right\}.$$

**Remark 1.** *It is easy to see that when $k = 0$, the Dunkl perimeter is reduced to the classical perimeter of $E \subseteq \Omega$, that is*
$$P_k(E, \Omega) = P(E, \Omega).$$

The following corollary can be easily obtained by replacing $f$ in Lemma 4 with $\chi_E$.

**Corollary 1.** *(Lower semicontinuity of Dunkl perimeter). Suppose $E, E_m \subset \Omega$, $m = 1, 2, \dots,$ then*

$$\liminf_{m \to \infty} P_k(E_m, \Omega) \geq P_k(E, \Omega).$$

From the definition of the Dunkl perimeter and plus the above max–min inequality, we have the following lemma.

**Lemma 9.** *If $E, F \subseteq \Omega$, then*

$$P_k(E \cap F, \Omega) + P_k(E \cup F, \Omega) \leq P_k(E, \Omega) + P_k(F, \Omega).$$

We give the following notation. For $f : \Omega \to \mathbb{R}$ and $t \in \mathbb{R}$, define

$$E_t := \{x \in \Omega \mid f(x) > t\}.$$

**Lemma 10.** *If $f \in BV_k(\Omega)$, the mapping*

$$t \mapsto \|\nabla_k \chi_{E_t}\|(\Omega) = P_k(E_t, \Omega)$$

*is Lebesgue measurable for $t \in \mathbb{R}$.*

**Theorem 1.** *Let $f \in BV_k(\Omega)$, then*

$$\|\nabla_k f\|(\Omega) \leq \int_{-\infty}^{+\infty} P_k(E_t, \Omega)\, dt. \tag{9}$$

**Proof.** Let $\varphi \in C_c^{\infty}(\Omega, \mathbb{R}^N)$ and $\|\varphi\|_{L^{\infty}} \leq 1$. Firstly, we prove the claim:

$$\int_{\Omega} f(x)\mathrm{div}_k \varphi(x)\, d\mu_k(x) = \int_{-\infty}^{+\infty} \left( \int_{E_t} \mathrm{div}_k \varphi(x)\, d\mu_k(x) \right) dt.$$

Suppose $f \geq 0$, we have

$$f(x) = \int_0^{\infty} \chi_{E_t}(x)\, dt$$

for *a.e.* $x \in \Omega$, and we obtain

$$
\begin{aligned}
\int_\Omega f(x)\mathrm{div}_k\varphi(x)\,d\mu_k(x) &= \int_\Omega \left(\int_0^\infty \chi_{E_t}(x)\,dt\right)\mathrm{div}_k\varphi(x)\,d\mu_k(x) \\
&= \int_0^\infty \left(\int_\Omega \chi_{E_t}(x)\mathrm{div}_k\varphi(x)\,d\mu_k(x)\right)dt \\
&= \int_0^\infty \left(\int_{\chi_{E_t}} \mathrm{div}_k\varphi(x)\,d\mu_k(x)\right)dt.
\end{aligned}
$$

Similarly, if $f \le 0$,

$$
f(x) = \int_{-\infty}^0 (\chi_{E_t}(x) - 1)\,dt,
$$

and we get

$$
\int_\Omega f(x)\mathrm{div}_k\varphi(x)\,d\mu_k(x) = \int_{-\infty}^0 \left(\int_{\chi_{E_t}} \mathrm{div}_k\varphi(x)\,d\mu_k(x)\right)dt.
$$

For the general case, write $f = f^+ + f^-$, therefore, we conclude that for all $\varphi$,

$$
\int_\Omega f(x)\mathrm{div}_k\varphi(x)\,d\mu_k(x) \le \int_{-\infty}^\infty P_k(E_t,\Omega)\,dt.
$$

Thus,

$$
\|\nabla_k f\|(\Omega) \le \int_{-\infty}^\infty P_k(E_t,\Omega)\,dt.
$$

□

The isoperimetric inequality for the $BV_k$ function is valid and it is proved in [28].

**Proposition 1.** *Let E be a bounded Lipschitz set of finite Dunkl perimeter on $\mathbb{R}^N$. Then*

$$
\mu_k(E)^{1-\frac{1}{N+2\gamma}} \le CP_k(E),
$$

*with the sharp constant $C = \frac{\mu_k(B_1^\epsilon)^{1-\frac{1}{N+2\gamma}}}{P(B_1^\epsilon)}$, where $\epsilon$ is any element of E such that $\mathbb{R}_\epsilon^N$ is a Weyl chamber, $B_1 = \{|x| < 1\}$ and $B_1^\epsilon = B_1 \cap \mathbb{R}_\epsilon^N$.*

Next we show that the Gauss–Green formula is valid on sets of locally finite Dunkl perimeter.

**Theorem 2.** *(Gauss–Green formula). Let E have locally finite perimeter. Assume E is G-invariant, that is $E = \{\sigma_\alpha x : x \in E, \alpha \in R_+\}$. Then*

$$
\int_E \mathrm{div}_k\varphi(x)\,d\mu_k(x) = \int_{\partial^*E} \varphi(x) \cdot \nu_E w_k(x)\,d\mathcal{H}^{N-1}(x), \tag{10}
$$

*for $\varphi \in C_c^\infty(\mathbb{R}^N, \mathbb{R}^N)$, where $\partial^*E = \{x : \nu_E \text{ exists}\}$ and the unit vector $\nu_E$ is the outward normal to E.*

**Proof.** Through calculation, we obtain

$$
\begin{aligned}
\int_E \mathrm{div}_k\varphi(x)\,d\mu_k(x) &= \int_E \mathrm{div}_k\varphi(x)w_k(x)\,dx \\
&= \int_E \left(\mathrm{div}\,\varphi(x) + \sum_{i=1}^N \sum_{\alpha\in R_+} k(\alpha)\alpha_i \frac{\varphi_i(x) - \varphi_i(\sigma_\alpha x)}{\langle \alpha, x\rangle}\right) w_k(x)\,dx \\
&= \int_E \mathrm{div}\,\varphi(x)w_k(x)\,dx + \int_E \sum_{i=1}^N \sum_{\alpha\in R_+} k(\alpha)\alpha_i \frac{\varphi_i(x) - \varphi_i(\sigma_\alpha x)}{\langle \alpha, x\rangle}w_k(x)\,dx \\
&= \int_{\partial^* E} \varphi(x)\cdot\nu_E w_k(x)\,d\mathcal{H}^{N-1}(x) - \int_E \varphi(x)\cdot\nabla w_k(x)\,dx \\
&\quad + \int_E \sum_{i=1}^N \sum_{\alpha\in R_+} k(\alpha)\alpha_i \frac{\varphi_i(x) - \varphi_i(\sigma_\alpha x)}{\langle \alpha, x\rangle}w_k(x)\,dx \\
&= \int_{\partial^* E} \varphi(x)\cdot\nu_E w_k(x)\,d\mathcal{H}^{N-1}(x) - \int_E \sum_{i=1}^N \sum_{\alpha\in R_+} 2k(\alpha)\alpha_i \frac{\varphi_i(x)}{\langle \alpha, x\rangle}w_k(x)\,dx \\
&\quad + \int_E \sum_{i=1}^N \sum_{\alpha\in R_+} k(\alpha)\alpha_i \frac{\varphi_i(x) - \varphi_i(\sigma_\alpha x)}{\langle \alpha, x\rangle}w_k(x)\,dx \\
&= \int_{\partial^* E} \varphi(x)\cdot\nu_E w_k(x)\,d\mathcal{H}^{N-1}(x) - \int_E \sum_{i=1}^N \sum_{\alpha\in R_+} k(\alpha)\alpha_i \frac{\varphi_i(x)}{\langle \alpha, x\rangle}w_k(x)\,dx \\
&\quad - \int_E \sum_{i=1}^N \sum_{\alpha\in R_+} k(\alpha)\alpha_i \frac{\varphi_i(\sigma_\alpha x)}{\langle \alpha, \sigma_\alpha x\rangle}w_k(\sigma_\alpha x)\,dx \\
&\quad + \int_E \sum_{i=1}^N \sum_{\alpha\in R_+} k(\alpha)\alpha_i \frac{\varphi_i(x) - \varphi_i(\sigma_\alpha x)}{\langle \alpha, x\rangle}w_k(x)\,dx \\
&= \int_{\partial^* E} \varphi(x)\cdot\nu_E w_k(x)\,d\mathcal{H}^{N-1}(x),
\end{aligned}
$$

where we have used the facts: $\langle \alpha, x\rangle = -\langle \alpha, \sigma_\alpha x\rangle$, $w_k(x) = w_k(\sigma_\alpha x)$, and the following facts for the derivatives of $w_k$:

$$
\partial_i |\langle \alpha, x\rangle|^{2k(\alpha)} = \begin{cases} 2k(\alpha)\alpha_i\langle \alpha, x\rangle^{2k(\alpha)-1} = 2k(\alpha)\alpha_i \frac{|\langle \alpha, x\rangle|^{2k(\alpha)}}{\langle \alpha, x\rangle}, & \text{if } \langle \alpha, x\rangle > 0, \\ -2k(\alpha)\alpha_i\langle \alpha, x\rangle^{2k(\alpha)-1} = 2k(\alpha)\alpha_i \frac{|\langle \alpha, x\rangle|^{2k(\alpha)}}{\langle \alpha, x\rangle}, & \text{if } \langle \alpha, x\rangle < 0. \end{cases}
$$

$\square$

## 3. Basic Facts of Dunkl BV Capacity

**Definition 3.** *For a set $E \subseteq \mathbb{R}^N$, let $\mathcal{A}\big(E, BV_k(\mathbb{R}^N)\big)$ be the class of admissible functions on $\mathbb{R}^N$, that is, functions $f \in BV_k(\mathbb{R}^N)$ satisfying $0 \le f \le 1$ and $f = 1$ in a neighborhood of $E$ (an open set containing $E$). The $BV_k$ capacity of $E$ is defined by*

$$
\mathrm{cap}\big(E, BV_k(\mathbb{R}^N)\big) := \inf\big\{\|\nabla_k f\|(\mathbb{R}^N) : f \in \mathcal{A}\big(E, BV_k(\mathbb{R}^N)\big)\big\}. \tag{11}
$$

Now we will see that capacity is partly suited for characterizing the partial properties of $BV_k$ functions.

**Theorem 3.** *If $E$ is an arbitrary subset of $\mathbb{R}^N$, then*

$$
\mathrm{cap}\big(E, BV_k(\mathbb{R}^N)\big) \le \inf_F P_k(F),
$$

*where the infimum is taken over all sets $F \subseteq \mathbb{R}^N$ such that $E \subseteq \mathrm{int}(F)$.*

**Proof.** If $E \subseteq \text{int}(F) \subseteq \mathbb{R}^N$ and $P_k(F) < \infty$, then

$$\chi_F \in \mathcal{A}(E, BV_k(\mathbb{R}^N)).$$

So

$$\text{cap}(E, BV_k(\mathbb{R}^N)) \leq P_k(F).$$

By taking the infimum over all such sets $F$, we obtain

$$\text{cap}(E, BV_k(\mathbb{R}^N)) \leq \inf_F P_k(F).$$

$\square$

**Theorem 4.** *(Measure theoretic properties of $BV_k$ capacity). The* $\text{cap}(\cdot, BV_k(\mathbb{R}^N))$ *enjoys the following properties.*

(i)
$$\text{cap}(\varnothing, BV_k(\mathbb{R}^N)) = 0.$$

(ii)  *Assume $E_1, E_2$ are subsets of $\mathbb{R}^N$. If $E_1 \subseteq E_2$, then*

$$\text{cap}(E_1, BV_k(\mathbb{R}^N)) \leq \text{cap}(E_2, BV_k(\mathbb{R}^N)).$$

(iii)  *If $E_1, E_2 \subseteq \mathbb{R}^N$, then*

$$\text{cap}(E_1 \cup E_2, BV_k(\mathbb{R}^N)) + \text{cap}(E_1 \cap E_2, BV_k(\mathbb{R}^N)) \leq \text{cap}(E_1, BV_k(\mathbb{R}^N)) + \text{cap}(E_2, BV_k(\mathbb{R}^N)).$$

*Especially, if $E_1 \subseteq E_2$ or $E_2 \subseteq E_1$, the equality holds.*

(iv)  *If $E_n, n = 1, 2, \ldots,$ are subsets of $\mathbb{R}^N$, then*

$$\text{cap}\left(\cup_{n=1}^{\infty} E_n, BV_k(\mathbb{R}^N)\right) \leq \sum_{n=1}^{\infty} \text{cap}(E_n, BV_k(\mathbb{R}^N)).$$

(v)
$$\lim_{n \to \infty} \text{cap}(E_n, BV_k(\mathbb{R}^N)) = \text{cap}\left(\cup_{n=1}^{\infty} E_n, BV_k(\mathbb{R}^N)\right)$$

*for any sequence $\{E_n\}_{n=1}^{\infty}$ of subsets of $\mathbb{R}^N$ with $E_1 \subseteq E_2 \subseteq E_3 \subseteq \ldots$.*

(vi)  *If $E_n, n = 1, 2, \ldots,$ are compact sets of $\mathbb{R}^N$ and $E_1 \supseteq E_2 \supseteq E_3 \supseteq \ldots$, then*

$$\lim_{n \to \infty} \text{cap}(E_n, BV_k(\mathbb{R}^N)) = \text{cap}\left(\cap_{n=1}^{\infty} E_n, BV_k(\mathbb{R}^N)\right).$$

(vii)  *For every $E \subseteq \mathbb{R}^N$, there has*

$$\text{cap}(E, BV_k(\mathbb{R}^N)) = \inf\left\{\text{cap}(O, BV_k(\mathbb{R}^N)) : \text{open } O \supseteq E\right\}.$$

(viii)  *For any Borel set $E \subseteq \mathbb{R}^N$ we have*

$$\text{cap}(E, BV_k(\mathbb{R}^N)) = \sup\left\{\text{cap}(K, BV_k(\mathbb{R}^N)) : \text{compact } K \subseteq E\right\}.$$

**Proof.**

(i), (ii). Assertions (i) and (ii) follow from (11).

(iii) Without losing generality, we can suppose

$$\text{cap}(E_1, BV_k(\mathbb{R}^N)) + \text{cap}(E_2, BV_k(\mathbb{R}^N)) < \infty.$$

For $\varepsilon > 0$, there exist two functions $f \in \mathcal{A}(E_1, BV_k(\mathbb{R}^N))$ and $g \in \mathcal{A}(E_2, BV_k(\mathbb{R}^N))$ such that

$$\begin{cases} \|\nabla_k f\|(\mathbb{R}^N) < \text{cap}(E_1, BV_k(\mathbb{R}^N)) + \frac{\varepsilon}{2}, \\ \|\nabla_k g\|(\mathbb{R}^N) < \text{cap}(E_2, BV_k(\mathbb{R}^N)) + \frac{\varepsilon}{2}. \end{cases}$$

Then, we get

$$\max\{f, g\} \in \mathcal{A}(E_1 \cup E_2, BV_k(\mathbb{R}^N)) \quad \text{and} \quad \min\{f, g\} \in \mathcal{A}(E_1 \cap E_2, BV_k(\mathbb{R}^N)),$$

and

$$\begin{aligned} &\text{cap}(E_1 \cup E_2, BV_k(\mathbb{R}^N)) + \text{cap}(E_1 \cap E_2, BV_k(\mathbb{R}^N)) \\ &\leq \|\nabla_k f\|(\mathbb{R}^N) + \|\nabla_k g\|(\mathbb{R}^N) \\ &\leq \text{cap}(E_1, BV_k(\mathbb{R}^N)) + \text{cap}(E_2, BV_k(\mathbb{R}^N)) + \varepsilon. \end{aligned}$$

Letting $\varepsilon \to 0$, the first statement is proved. Especially, when $E_1 \subseteq E_2$ or $E_2 \subseteq E_1$, we can get

$$\text{cap}(E_1 \cup E_2, BV_k(\mathbb{R}^N)) + \text{cap}(E_1 \cap E_2, BV_k(\mathbb{R}^N)) = \text{cap}(E_1, BV_k(\mathbb{R}^N)) + \text{cap}(E_2, BV_k(\mathbb{R}^N)).$$

(iv) Assume

$$\sum_{n=1}^{\infty} \text{cap}(E_n, BV_k(\mathbb{R}^N)) < \infty.$$

For any $\varepsilon > 0$ and $n = 1, 2, \ldots$, there is

$$f_n \in \mathcal{A}(E_n, BV_k(\mathbb{R}^N))$$

so that

$$\|\nabla_k f_n\|(\mathbb{R}^N) < \text{cap}(E_n, BV_k(\mathbb{R}^N)) + \frac{\varepsilon}{2^n}.$$

Upon setting $f = \sup_n f_n$, we have

$$\|\nabla_k f\|(\mathbb{R}^N) \leq \sum_{n=1}^{\infty} \|\nabla_k f_n\|(\mathbb{R}^N) < \sum_{n=1}^{\infty} \text{cap}(E_n, BV_k(\mathbb{R}^N)) + \varepsilon < \infty,$$

which implies $f \in \mathcal{A}(\cup_{n=1}^{\infty} E_n, BV_k(\mathbb{R}^N))$. Thus, we get

$$\begin{aligned} \|\nabla_k f\|(\mathbb{R}^N) &\leq \liminf_{n \to \infty} \|\nabla_k \max\{f_1, \ldots, f_n\}\|(\mathbb{R}^N) \\ &\leq \sum_{n=1}^{\infty} \|\nabla_k f_n\|(\mathbb{R}^N) \\ &\leq \sum_{n=1}^{\infty} \text{cap}(E_n, BV_k(\mathbb{R}^N)) + \varepsilon. \end{aligned}$$

Letting $\varepsilon \to 0$, we complete the countable subadditivity of (iv).

(v) Suppose $\{E_n\}_{n=1}^{\infty}$ is an increasing sequence. It is easy to observe that

$$\lim_{n \to \infty} \text{cap}(E_n, BV_k(\mathbb{R}^N)) \leq \text{cap}(\cup_{n=1}^{\infty} E_n, BV_k(\mathbb{R}^N)).$$

Next we just consider when

$$\lim_{n \to \infty} \text{cap}(E_n, BV_k(\mathbb{R}^N)) < \infty,$$

the equality holds. Let $\varepsilon > 0$ and assume

$$\lim_{n \to \infty} \text{cap}(E_n, BV_k(\mathbb{R}^N)) < \infty.$$

For $n = 1, 2, \ldots$, there is

$$f_n \in \mathcal{A}\big(E_n, BV_k(\mathbb{R}^N)\big)$$

satisfying

$$\|\nabla_k f_n\|(\mathbb{R}^N) < \mathrm{cap}\big(E_n, BV_k(\mathbb{R}^N)\big) + \frac{\varepsilon}{2^n}.$$

We consider

$$\begin{cases} g_i = \max_{1 \le i \le n} f_i = \max\{g_{i-1}, f_i\}; \\ g_0 = 0; \\ E_0 = \varnothing; \\ h_i = \min\{g_{i-1}, f_i\}. \end{cases}$$

Then we can get $g_i, h_i \in BV_k(\mathbb{R}^N)$ and $E_n \subseteq \mathrm{int}\{x \in \mathbb{R}^N : h_{n+1}(x) = 1\}$.
Since $g_i = \max\{g_{i-1}, g_i\}$ and Lemma 7, we derive that

$$\|\nabla_k \max\{g_{i-1}, g_i\}\|(\mathbb{R}^N) + \|\nabla_k \min\{g_{i-1}, g_i\}\|(\mathbb{R}^N) \le \|\nabla_k g_{i-1}\|(\mathbb{R}^N) + \|\nabla_k g_i\|(\mathbb{R}^N),$$

and thereby using $E_n \subseteq E_{n+1}$ to achieve

$$\begin{aligned} \|\nabla_k g_i\|(\mathbb{R}^N) + \mathrm{cap}\big(E_{i-1}, BV_k(\mathbb{R}^N)\big) & \\ &\le \|\nabla_k g_i\|(\mathbb{R}^N) + \|\nabla_k h_i\|(\mathbb{R}^N) \\ &\le \|\nabla_k g_i\|(\mathbb{R}^N) + \|\nabla_k g_{i-1}\|(\mathbb{R}^N) \\ &\le \|\nabla_k g_{i-1}\|(\mathbb{R}^N) + \mathrm{cap}\big(E_i, BV_k(\mathbb{R}^N)\big) + \frac{\varepsilon}{2^i}, \end{aligned}$$

therefore,

$$\begin{aligned} \|\nabla_k g_i\|(\mathbb{R}^N) - \|\nabla_k g_{i-1}\|(\mathbb{R}^N) & \\ &\le \mathrm{cap}\big(E_i, BV_k(\mathbb{R}^N)\big) - \mathrm{cap}\big(E_{i-1}, BV_k(\mathbb{R}^N)\big) + \frac{\varepsilon}{2^i}. \end{aligned}$$

By adding the above inequalities, we get

$$\|\nabla_k g_n\|(\mathbb{R}^N) \le \mathrm{cap}\big(E_n, BV_k(\mathbb{R}^N)\big) + \varepsilon.$$

So, let $f = \lim_{n \to \infty} g_n$. Using the monotone convergence theorem, we own

$$\|\nabla_k f\|(\mathbb{R}^N) \le \lim_{n \to \infty} \|\nabla_k g_n\|(\mathbb{R}^N) \le \lim_{n \to \infty} \mathrm{cap}\big(E_n, BV_k(\mathbb{R}^N)\big) + \varepsilon.$$

Then via the lower semicontinuity of the Dunkl variation, we have

$$f \in \mathcal{A}\big(\cup_{n=1}^{\infty} E_n, BV_k(\mathbb{R}^N)\big),$$

and so

$$\begin{aligned} \mathrm{cap}\big(\cup_{n=1}^{\infty} E_n, BV_k(\mathbb{R}^N)\big) &\le \|\nabla_k f\|(\mathbb{R}^N) \\ &\le \liminf_{n \to \infty} \|\nabla_k g_n\|(\mathbb{R}^N) \\ &\le \lim_{n \to \infty} \mathrm{cap}\big(E_n, BV_k(\mathbb{R}^N)\big) + \varepsilon. \end{aligned}$$

(vi) If $E_n, n = 1, 2, \ldots$, are compact sets of $\mathbb{R}^N$ and $E_1 \supseteq E_2 \supseteq E_3 \supseteq \ldots$. Let $E = \cap_{n=1}^{\infty} E_n$ and notice that it is also a compact set. For any $0 < \varepsilon < \frac{1}{2}$, there has

$$f \in \mathcal{A}\big(E, BV_k(\mathbb{R}^N)\big)$$

so that

$$\|\nabla_k f\|(\mathbb{R}^N) < \mathrm{cap}\big(E, BV_k(\mathbb{R}^N)\big) + \varepsilon.$$

If $n$ is large enough, then $E_n$ is contained in

$$\{x \in \mathbb{R}^N : f(x) \geq 1 - \frac{\varepsilon}{2}\} \subseteq \mathrm{int}\{x \in \mathbb{R}^N : f(x) \geq 1 - \varepsilon\},$$

thus

$$\min\{1, \frac{1}{1 - \varepsilon}f\} \in \mathcal{A}(\{x \in \mathbb{R}^N : f(x) \geq 1 - \frac{\varepsilon}{2}\}, BV_k(\mathbb{R}^N)).$$

Applying (ii) we immediately get

$$\lim_{n \to \infty} \mathrm{cap}(E_n, BV_k(\mathbb{R}^N)) \leq \mathrm{cap}(\{x \in \mathbb{R}^N : f(x) \geq 1 - \varepsilon\}, BV_k(\mathbb{R}^N))$$

$$\leq \frac{1}{1 - \varepsilon}\|\nabla_k f\|(\mathbb{R}^N)$$

$$\leq \frac{1}{1 - \varepsilon}\Big(\mathrm{cap}(E, BV_k(\mathbb{R}^N)) + \varepsilon\Big).$$

Letting $\varepsilon \to 0$, we can get

$$\mathrm{cap}(E, BV_k(\mathbb{R}^N)) \leq \lim_{n \to \infty} \mathrm{cap}(E_n, BV_k(\mathbb{R}^N)) \leq \mathrm{cap}(E, BV_k(\mathbb{R}^N)).$$

(vii) From (ii), we know that

$$\mathrm{cap}(E, BV_k(\mathbb{R}^N)) \leq \inf\{\mathrm{cap}(O, BV_k(\mathbb{R}^N)) : \mathrm{open}\ O \supseteq E\}.$$

To show the opposite inequality, we can suppose

$$\mathrm{cap}(E, BV_k(\mathbb{R}^N)) < \infty.$$

From (13), and for any $\varepsilon > 0$, there has $f \in \mathcal{A}(E, BV_k(\mathbb{R}^N))$ such that

$$\|\nabla_k f\|(\mathbb{R}^N) < \mathrm{cap}(E, BV_k(\mathbb{R}^N)) + \varepsilon.$$

Thus, there exists an open set $O \supseteq E$ such that $f = 1$ on $O$ and

$$\mathrm{cap}(O, BV_k(\mathbb{R}^N)) \leq \|\nabla_k f\|(\mathbb{R}^N) < \mathrm{cap}(E, BV_k(\mathbb{R}^N)) + \varepsilon.$$

Consequently,

$$\inf\{\mathrm{cap}(O, BV_k(\mathbb{R}^N)) : \mathrm{open}\ O \supseteq E\} \leq \mathrm{cap}(E, BV_k(\mathbb{R}^N)).$$

(viii) This claim can be obtained from (v) and (vi).  □

We learn from [29] and introduce the following notion and some results which are similar to various spaces, such as [18,19].

**Definition 4.** *Suppose $E \subset \mathbb{R}^N$ and $\varepsilon > 0$, let*

$$\mathcal{H}_{\varepsilon,k}(E) := \inf\left\{\sum_{i=1}^{\infty} r_i^{-1}\mu_k(B(x_i, r_i)) : E \subseteq \cup_{i=1}^{\infty}B(x_i, r_i) \text{ with } 0 < r_i \leq \varepsilon\right\},$$

*and we call*

$$\mathcal{H}_k(E) := \lim_{\varepsilon \to 0} \mathcal{H}_{\varepsilon,k}(E)$$

*is the Dunkl Hausdorff measure of codimension one of E.*

Of course, when $k = 0$, it becomes the classical Hausdorff measure of codimension one.

**Theorem 5.** *If $E$ is a Borel subset of $\mathbb{R}^N$, then*

$$\mathcal{H}_k(E) = 0 \Rightarrow \mathrm{cap}\big(E, BV_k(\mathbb{R}^N)\big) = 0.$$

**Proof.** If $\mathcal{H}_k(E) = 0$ and $0 < \varepsilon < 1$, we consider a ball-cover $\big\{B(x_i, r_i)\big\}_{i \in \mathbb{N}}$ that

$$\begin{cases} x_i \in E; \\ r_i \in (0,1); \\ \sum_{i=1}^\infty r_i^{-1} \mu_k\big(B(x_i, r_i)\big) < \varepsilon. \end{cases}$$

Let $f_i(x) = \max\{0, 1 - r_i^{-1}|x - x_i|\}$, we have

$$\begin{aligned}
\mathrm{cap}\big(B(x_i, r_i), BV_k(\mathbb{R}^N)\big) &\leq \int_{\mathbb{R}^N} |\nabla_k f_i(x)|\, d\mu_k(x) \\
&\leq \int_{B(x_i, r_i)} |\nabla f_i(x)|\, d\mu_k(x) \\
&\quad + \int_{B(x_i, r_i)} \Big| \sum_{\alpha \in R_+} k(\alpha)|\alpha_i| \frac{|x - x_i| - |\sigma_\alpha(x) - x_i|}{r_i \cdot |\langle \alpha, x \rangle|} \Big|\, d\mu_k(x) \\
&\leq \Big( 1 + \sum_{\alpha \in R_+} k(\alpha)|\alpha_i|\alpha \Big) r_i^{-1} \mu_k\big(B(x_i, r_i)\big),
\end{aligned}$$

then

$$\mathrm{cap}\big(E, BV_k(\mathbb{R}^N)\big) \leq \sum_{i=1}^\infty \mathrm{cap}\big(B(x_i, r_i), BV_k(\mathbb{R}^N)\big) \leq \Big( 1 + \sum_{\alpha \in R_+} k(\alpha)|\alpha_i|\alpha \Big) \varepsilon.$$

By taking $\varepsilon \to 0$, it follows that

$$\mathrm{cap}\big(E, BV_k(\mathbb{R}^N)\big) = 0.$$

$\square$

## 4. Heat Semigroups Characterization of Dunkl Bounded Variation Functions

At first, we recall the Dunkl heat kernel and collect its properties established by Rösler (cf. [24,26]). One important function is the Dunkl kernel $E(x, y)$ associated with Dunkl operators. For generic multiplicities $k$ and each $y \in \mathbb{R}^N$, the system

$$\begin{cases} T_i E(x, y) = y_i E(x, y) & i = 1, 2, \ldots, N \\ E(0, y) = 1 \end{cases} \tag{12}$$

has a unique solution on $\mathbb{R}^N$, which generalizes the exponential functions $e^{\langle x, y \rangle}$. For functions of non-negative multiplicity, the commutative algebra of Dunkl operators and the algebra of general partial differential operators are intertwined by a unique linear homogeneous isomorphism over polynomials. In other words, there exists a unique intertwining operator $V_k$ such that

$$T_i V_k = V_k \partial_i,$$

which also justifies the next formula

$$E(x, y) = V_k(e^{\langle \cdot, y \rangle})(x), \quad x \in \mathbb{R}^N.$$

Note that the Dunkl kernel $E(x, y)$ is an explicit "closed" form, where it is known so far only in some particular cases. In dimension 1, the Dunkl kernel can be written as a sum of two Bessel functions (cf. [24])

$$E(x, y) = j_{k-1/2}(xy) + \frac{xy}{2k+1} j_{k+1/2}(xy).$$

The generalized Bessel function is written as

$$J(x, y) := \frac{1}{|G|} \sum_{g \in G} E(gx, y).$$

Here $|G|$ is the order of the group $G$. In dimension 1,

$$J(x, y) = j_{k-1/2}(xy).$$

Let us collect the main properties of Dunkl kernel in the following proposition (cf. [24,26]).

**Proposition 2.** *(Properties of the Dunkl kernel).*

- $E(x, y) = E(y, x)$;
- $E(gx, gy) = E(x, y) \quad \forall g \in G$;
- $E(\lambda x, y) = E(x, \lambda y) \quad \forall \lambda \in \mathbb{R}$;
- $\overline{E(x, y)} = E(\bar{x}, \bar{y})$;
- $0 < E(x, y) \le e^{\langle x, y \rangle}$;
- $|\partial_y^\beta E(x, y)| \le |x|^{|\beta|} \max_{g \in G} e^{\langle gx, y \rangle}$. Specially, $|E(-ix, y)| \le 1$.

In [26], a Dunkl transform is given by

$$\mathcal{H}f(x) = \int_{\mathbb{R}^N} f(y) E(-ix, y) \, d\mu_k(y). \tag{13}$$

The Dunkl heat equation

$$\begin{cases} \partial_t H_t f(x) = \Delta_x H_t f(x) \\ H_0 f(x) = f(x) \end{cases}$$

can be obtained via the Dunkl transform (13) (see Theorem 3.12 in [26] for more details) under suitable cases, where

$$H_t f(x) := \int_{\mathbb{R}^N} f(y) h_t(x, y) \, d\mu_k(y),$$

and the Dunkl heat kernel is defined by

$$h_t(x, y) = M_k^{-1} (2t)^{-(\gamma + N/2)} e^{-(\frac{|x|^2 + |y|^2}{4t})} E(\frac{x}{\sqrt{2t}}, \frac{y}{\sqrt{2t}}) \quad \forall t > 0, \forall x, y \in \mathbb{R}^N.$$

Here $M_k$ is Macdonald-Mehta integral associated with the root system $R$ and it is represented as

$$M_k = \int_{\mathbb{R}^N} e^{-\frac{|x|^2}{2}} \, d\mu_k(x).$$

Next, we collect a series of basic properties of the Dunkl heat kernel (cf. [24,26]).

**Proposition 3.** *(Basic properties of the Dunkl heat kernel).*

- $h_t(x, y)$ is an analytic function in $(t, x, y) \in (0, +\infty) \times \mathbb{R}^N \times \mathbb{R}^N$;
- $h_t(x, y) = h_t(y, x)$;

- $h_t(x, y) > 0$ and $\int_{\mathbb{R}^N} h_t(x, y)\, d\mu_k(y) = 1$;
- $h_{s+t}(x, y) = \int_{\mathbb{R}^N} h_s(x, z)h_t(y, z)\, d\mu_k(z)$;
- $h_t(x, y) \leq M_k^{-1}(2t)^{-(\gamma+N/2)} \max_{g \in G} e^{-\frac{|gx-y|^2}{4t}}$.

**Lemma 11.** *The Dunkl semigroup* $\{H_t\}_{t \geq 0}$ *is symmetric in* $L^2(\mu_k)$.

**Proof.** We obtain

$$
\begin{aligned}
\int_{\mathbb{R}^N} H_t f(x)g(x)\, du_k(x) &= \int_{\mathbb{R}^N} \int_{\mathbb{R}^N} h_t(x, y)f(y)\, du_k(y)g(x)\, du_k(x) \\
&= \int_{\mathbb{R}^N} \int_{\mathbb{R}^N} h_t(x, y)f(y)g(x)\, du_k(y)\, du_k(x) \\
&= \int_{\mathbb{R}^N} \int_{\mathbb{R}^N} h_t(y, x)g(x)\, du_k(x)f(y)\, du_k(y) \\
&= \int_{\mathbb{R}^N} H_t g(x)f(x)\, du_k(x),
\end{aligned}
$$

which completes the proof.  □

**Lemma 12.** *For every* $f(x) \in L^1(\mu_k)$, *we have*

$$
\lim_{t \to 0} H_t f(x) = f(x) \quad \text{in } L^1(\mu_k).
$$

**Proof.** Firstly, define

$$
\rho(x) := M_k^{-1} \int_{\mathbb{R}^N} e^{-\frac{|x|^2+|y|^2}{2}} E(x, y)\, d\mu_k(y).
$$

It's clear from the calculation that $\rho(0) = 1$. When $x \neq 0$, according to the fact: the function $e^{-\frac{|x|^2+|y|^2}{2}}$ is $G-$invariant, the Formulas (3) and (12), we have

$$
\begin{aligned}
0 = T_i^y \rho(x) &= T_i^y M_k^{-1} \int_{\mathbb{R}^N} e^{-\frac{|x|^2+|y|^2}{2}} E(x, y)\, d\mu_k(y) \\
&= M_k^{-1} \int_{\mathbb{R}^N} T_i^y e^{-\frac{|x|^2+|y|^2}{2}} \cdot E(x, y) + e^{-\frac{|x|^2+|y|^2}{2}} \cdot T_i^y E(x, y)\, d\mu_k(y) \\
&= M_k^{-1} \int_{\mathbb{R}^N} -y_i e^{-\frac{|x|^2+|y|^2}{2}} \cdot E(x, y) + e^{-\frac{|x|^2+|y|^2}{2}} \cdot x_i E(x, y)\, d\mu_k(y),
\end{aligned}
$$

where $T_i^y$ denotes the Dunkl directional derivative of the variable $y$. Similarly,

$$
\begin{aligned}
T_i \rho(x) &= T_i M_k^{-1} \int_{\mathbb{R}^N} e^{-\frac{|x|^2+|y|^2}{2}} E(x, y)\, d\mu_k(y) \\
&= M_k^{-1} \int_{\mathbb{R}^N} -x_i e^{-\frac{|x|^2+|y|^2}{2}} \cdot E(x, y) + e^{-\frac{|x|^2+|y|^2}{2}} \cdot y_i E(x, y)\, d\mu_k(y) \\
&= -T_i^y \rho(x) \\
&= 0.
\end{aligned}
$$

Thus, we get

$$
\rho(x) = \rho(0) \equiv 1. \tag{14}
$$

Following the definition of $H_t f(x)$ and (14), we obtain

$$\int_{\mathbb{R}^N} |H_t f(x) - f(x)| \, d\mu_k(x)$$

$$= \int_{\mathbb{R}^N} \left| \int_{\mathbb{R}^N} f(y) h_t(x,y) \, d\mu_k(y) - f(x) \int_{\mathbb{R}^N} h_t(x,y) \, d\mu_k(y) \right| d\mu_k(x)$$

$$= \int_{\mathbb{R}^N} \left| \int_{\mathbb{R}^N} h_t(x,y)[f(x) - f(y)] \, d\mu_k(y) \right| d\mu_k(x)$$

$$= \int_{\mathbb{R}^N} \left| \int_{\mathbb{R}^N} M_k^{-1}(2t)^{-(\gamma+N/2)} e^{-\frac{|x|^2+|y|^2}{4t}} E\left(\frac{x}{\sqrt{2t}}, \frac{y}{\sqrt{2t}}\right)[f(x) - f(y)] \, d\mu_k(y) \right| d\mu_k(x)$$

$$= \int_{\mathbb{R}^N} \left| \int_{\mathbb{R}^N} M_k^{-1} e^{-\frac{|x|^2+|y|^2}{2}} E(x,y)\left[f(\sqrt{2t}x) - f(\sqrt{2t}y)\right] d\mu_k(y) \right| d\mu_k(x).$$

Letting $t \to 0$, via the dominated convergence theorem we prove that

$$\lim_{t \to 0} \|H_t f(x) - f(x)\|_{L^1(\mu_k)} = 0.$$

□

**Lemma 13.** *The Dunkl semigroup* $\{H_t\}_{t \in [0,+\infty)}$ *satisfies the following properties:*

(i)    $t \mapsto H_t f$ *is continuous from* $[0, \infty)$ *to* $L^2(\mu_k)$.

(ii)    $|\nabla_k H_t f(x)| = \left| \int_{\mathbb{R}^N} h_t(x,y) \nabla_k^y f(y) \, d\mu_k(y) \right|$, *where* $\nabla_k^y$ *denotes the Dunkl gradient of the variable* $y$, $i = 1, \ldots, N$.

(iii)    $\|H_t f\|_{L^\infty} \le \|f\|_{L^\infty}$    $\forall f \in C_c^\infty(\mathbb{R}^N)$.

**Proof.** The property (i) is obviously available. Next we prove (ii). Firstly, since the function $e^{\frac{-(|x|^2+|y|^2)}{4t}}$ is $G$-invariant, we get

$$T_i h_t(x,y) = T_i M_k^{-1}(2t)^{-(\gamma+N/2)} e^{-\left(\frac{|x|^2+|y|^2}{4t}\right)} E\left(\frac{x}{\sqrt{2t}}, \frac{y}{\sqrt{2t}}\right)$$

$$= M_k^{-1}(2t)^{-(\gamma+N/2)} e^{-\left(\frac{|x|^2+|y|^2}{4t}\right)} \left(-\frac{x_i}{2t}\right) E\left(\frac{x}{\sqrt{2t}}, \frac{y}{\sqrt{2t}}\right)$$

$$+ M_k^{-1}(2t)^{-(\gamma+N/2)} e^{-\left(\frac{|x|^2+|y|^2}{4t}\right)} \left(\frac{y_i}{\sqrt{2t}}\right) \frac{1}{\sqrt{2t}} E\left(\frac{x}{\sqrt{2t}}, \frac{y}{\sqrt{2t}}\right)$$

$$= \left(\frac{y_i - x_i}{2t}\right) M_k^{-1}(2t)^{-(\gamma+N/2)} e^{-\left(\frac{|x|^2+|y|^2}{4t}\right)} E\left(\frac{x}{\sqrt{2t}}, \frac{y}{\sqrt{2t}}\right).$$

By the same calculation, we have

$$T_i^y h_t(x,y) = \left(\frac{x_i - y_i}{2t}\right) M_k^{-1}(2t)^{-(\gamma+N/2)} e^{-\left(\frac{|x|^2+|y|^2}{4t}\right)} E\left(\frac{x}{\sqrt{2t}}, \frac{y}{\sqrt{2t}}\right).$$

So we obtain

$$\nabla_k h_t(x,y) = -\nabla_k^y h_t(x,y).$$

Next via the definition of $H_t f(x)$ and the property of the Dunkl gradient we have

$$
\begin{aligned}
\nabla_k H_t f(x) &= \nabla_k \int_{\mathbb{R}^N} h_t(x,y) f(y) \, d\mu_k(y) \\
&= \int_{\mathbb{R}^N} \nabla_k h_t(x,y) f(y) \, d\mu_k(y) \\
&= - \int_{\mathbb{R}^N} \nabla_k^y h_t(x,y) f(y) \, d\mu_k(y) \\
&= \int_{\mathbb{R}^N} h_t(x,y) \nabla_k^y f(y) \, d\mu_k(y).
\end{aligned}
$$

For (iii), it is easy to see that

$$
|H_t f(x)| = \left| \int_{\mathbb{R}^N} h_t(x,y) f(y) \, d\mu_k(y) \right| \leq \int_{\mathbb{R}^N} h_t(x,y) \, d\mu_k(y) \cdot \|f\|_{L^\infty},
$$

and then we take the infinite norm on both sides to get

$$
\|H_t f(x)\|_{L^\infty} \leq \left\| \int_{\mathbb{R}^N} h_t(x,y) \, d\mu_k(y) \right\|_{L^\infty} \cdot \|f\|_{L^\infty} \leq \|f\|_{L^\infty}.
$$

$\square$

**Theorem 6.** *Denote by $C_{bd}^\infty(\mathbb{R}^N, \mathbb{R}^N)$ the space of vector-valued functions with continuous partial derivatives and bounded Dunkl divergence. For each $f \in L^1(\mu_k)$, we have*

$$
\|\nabla_k f\| = \sup \left\{ \int_{\mathbb{R}^N} f(x) \mathrm{div}_k \varphi(x) \, d\mu_k(x) : \ \varphi \in C_{bd}^\infty(\mathbb{R}^N, \mathbb{R}^N), \ \|\varphi\|_{L^\infty} \leq 1 \right\}.
$$

**Proof.** Firstly, we can easily get

$$
\|\nabla_k f\| \leq \sup \left\{ \int_{\mathbb{R}^N} f(x) \mathrm{div}_k \varphi(x) \, d\mu_k(x) : \varphi \in C_{bd}^\infty(\mathbb{R}^N, \mathbb{R}^N), \ \|\varphi\|_{L^\infty} \leq 1 \right\}.
$$

Next we mainly prove the opposite inequality, let $\phi_n$ be a sequence of functions such that
(i)     $\phi_n$ is $G$-invariant and $0 \leq \phi_n \leq 1$ for all $x \in \mathbb{R}^N$ and $k \in \mathbb{N}$;
(ii)    there exists $n_K$ such that $\phi_n = 1$ on $\mathbb{R}^N$ for every compact set $K \subset \mathbb{R}^N$, $n \geq n_K$;
(iii)   $\|\nabla_k \phi_n\|_{L^\infty} \to 0$ as $n \to \infty$.
If $\varphi \in C_{bd}^\infty(\mathbb{R}^N, \mathbb{R}^N)$, we have $\|\phi_n \varphi\|_{L^\infty} \leq \|\varphi\|_{L^\infty}$, and

$$
\begin{aligned}
\mathrm{div}_k(\varphi \phi_n) &= \sum_{i=1}^N T_i(\varphi \phi_n) \\
&= \sum_{i=1}^N [T_i(\varphi) \phi_n + \varphi T_i(\phi_n)] \\
&= \sum_{i=1}^N \left[ \left( \partial_i(\varphi) + \sum_{\alpha \in R_+} k(\alpha) \alpha_i \frac{\varphi - \varphi(\sigma_\alpha x)}{\langle \alpha, x \rangle} \right) \phi_n + \varphi \partial_i(\phi_n) \right] \\
&= \mathrm{div}(\varphi \phi_n) + \sum_{i=1}^N \sum_{\alpha \in R_+} k(\alpha) \alpha_i \frac{\varphi - \varphi(\sigma_\alpha x)}{\langle \alpha, x \rangle} \phi_n \\
&= \phi_n \mathrm{div}\varphi + \sum_{i=1}^N \sum_{\alpha \in R_+} k(\alpha) \alpha_i \frac{\varphi - \varphi(\sigma_\alpha x)}{\langle \alpha, x \rangle} \phi_n + \varphi \cdot \nabla \phi_n \\
&= \phi_n \mathrm{div}_k \varphi + \varphi \cdot \nabla \phi_n.
\end{aligned}
$$

Therefore, if $\varphi \in C_{bd}^{\infty}(\mathbb{R}^N, \mathbb{R}^N)$ and $\|\varphi\|_{L^\infty} \leq 1$, then we get the following inequality by the dominated convergence theorem,

$$\int_{\mathbb{R}^N} f(x) \text{div}_k \varphi(x) \, d\mu_k(x) = \lim_{n \to \infty} \int_{\mathbb{R}^N} f(x) \text{div}_k(\phi_n(x) \varphi(x)) \, d\mu_k(x) \leq \|\nabla_k f\|.$$

Thus, the proof of Theorem 6 is completed. $\square$

**Example 1.** *We give an example for $\phi_n$ in the proof of Theorem 6 as follows:*

$$\phi_n(x) = \begin{cases} 1, & \text{if } |x| < n, \\ e^{\frac{1}{(|x|-n)^2-1}+1}, & \text{if } n \leq |x| \leq n+1, \\ 0, & \text{if } |x| > n+1. \end{cases}$$

*Note that the example we constructed implies that $\phi_n(x)$ exists for $n = 1, 2, \ldots$, and it satisfies the above conditions (i)–(iii) in Theorem 6. It is easy to see that (i) and (ii) hold true. Since $\phi_n(x)$ is G-invariant, we have*

$$\partial_i \phi_n(x) = \begin{cases} \frac{2(|x|-n)x_i}{[(|x|-n)^2-1]^2|x|} \cdot e^{\frac{1}{(|x|-n)^2-1}}, & \text{if } n \leq |x| \leq n+1, \\ 0, & \text{otherwise.} \end{cases}$$

*Thus, $\|\nabla_k \phi_n\|_{L^\infty} \to 0$ as $n \to \infty$, which satisfies (iii).*

**Theorem 7.** *For any $f(x) \in L^1(\mu_k)$, we have*

$$\|\nabla_k f\| = \lim_{t \to 0} \|\nabla_k H_t f(x)\|_{L^1(u_k)}.$$

**Proof.** At first, for every functions $f(x) \in BV_k(\mathbb{R}^N)$ and $\varphi \in C_c^\infty(\mathbb{R}^N, \mathbb{R}^N)$, we note that

$$\int_{\mathbb{R}^N} \nabla_k f(x) \cdot \varphi(x) \, d\mu_k(x) = - \int_{\mathbb{R}^N} f(x) \text{div}_k \varphi(x) \, d\mu_k(x).$$

Via the definition of $\|\nabla_k f\|$, Lemmas 12 and 13, we have

$$\begin{aligned} \int_{\mathbb{R}^N} f(x) \text{div}_k \varphi(x) \, d\mu_k(x) &= \lim_{t \to 0} \int_{\mathbb{R}^N} H_t f(x) \text{div}_k \varphi(x) \, d\mu_k(x) \\ &= -\lim_{t \to 0} \int_{\mathbb{R}^N} \nabla_k H_t f(x) \cdot \varphi(x) \, d\mu_k(x) \\ &\leq \lim_{t \to 0} \|\nabla_k H_t f(x)\|_{L^1(\mu_k)}. \end{aligned}$$

Then taking the supremum over $\varphi$, we get that

$$\|\nabla_k f\| \leq \lim_{t \to 0} \|\nabla_k H_t f(x)\|_{L^1(\mu_k)}.$$

Next, we prove the opposite inequality

$$\|\nabla_k f\| \geq \lim_{t \to 0} \|\nabla_k H_t f(x)\|_{L^1(\mu_k)}.$$

Let $\varphi$ be a vector function in $C_c^\infty(\mathbb{R}^N, \mathbb{R}^N)$ such that $\|\varphi\|_{L^\infty} \leq 1$. Denote by $H_t\varphi(x) := \int_{\mathbb{R}^N} h_t(x,y)\varphi(y)\, d\mu_k(y)$. We next explain that $H_t\varphi(x) \in C_{bd}^\infty(\mathbb{R}^N, \mathbb{R}^N)$.

$$
\|\mathrm{div}_k H_t\varphi(x)\|_{L^\infty} = \left\|\mathrm{div}_k \int_{\mathbb{R}^N} h_t(x,y)\varphi(y)\, d\mu_k(y)\right\|_{L^\infty}
$$
$$
= \left\|\int_{\mathbb{R}^N} h_t(x,y)\mathrm{div}_k\varphi(y)\, d\mu_k(y)\right\|_{L^\infty}
$$
$$
\leq \|\mathrm{div}_k\varphi\|_{L^\infty} < \infty,
$$

where we have used the fact that $\|H_t\varphi(x)\|_{L^\infty} \leq 1$ and Lemma 13 (ii). Therefore, we get

$$
\left|\int_{\mathbb{R}^N} \nabla_k H_t f(x) \cdot \varphi(x)\, d\mu_k(x)\right| = \left|\int_{\mathbb{R}^N} \nabla_k \int_{\mathbb{R}^N} h_t(x,y) f(y)\, d\mu_k(y) \cdot \varphi(x)\, d\mu_k(x)\right|
$$
$$
= \left|\int_{\mathbb{R}^N} \int_{\mathbb{R}^N} \nabla_k h_t(x,y) f(y)\, d\mu_k(y) \cdot \varphi(x)\, d\mu_k(x)\right|
$$
$$
= \left|\int_{\mathbb{R}^N} \int_{\mathbb{R}^N} -\nabla_k^y h_t(x,y) f(y)\, d\mu_k(y) \cdot \varphi(x)\, d\mu_k(x)\right|
$$
$$
= \left|\int_{\mathbb{R}^N} \int_{\mathbb{R}^N} h_t(x,y) \nabla_k^y f(y)\, d\mu_k(y) \cdot \varphi(x)\, d\mu_k(x)\right|
$$
$$
= \left|\int_{\mathbb{R}^N} \int_{\mathbb{R}^N} \nabla_k f(x) h_t(y,x) \cdot \varphi(y)\, d\mu_k(x)\, d\mu_k(y)\right|
$$
$$
= \left|\int_{\mathbb{R}^N} \nabla_k f(x) \cdot \int_{\mathbb{R}^N} h_t(x,y)\varphi(y)\, d\mu_k(y)\, d\mu_k(x)\right|
$$
$$
= \left|\int_{\mathbb{R}^N} f(x) \cdot \mathrm{div}_k H_t\varphi(x)\, d\mu_k(x)\right|
$$
$$
\leq \|\nabla_k f\|,
$$

where we have used the property that Dunkl semigroup $\{H_t\}_{t\geq 0}$ is symmetric in $L^2(\mu_k)$ and Theorem 6. Taking the supremum over all $\varphi \in C_c^\infty(\mathbb{R}^N, \mathbb{R}^N)$ and $\|\varphi\|_{L^\infty} \leq 1$ yield the desired inequality. Finally, letting $t \to 0$, we complete the proof. $\square$

## 5. Conclusions

In conclusion, this paper introduced and studied functions of Dunkl-bounded variation on $\mathbb{R}^N$. It was not obvious that results of the classical BV functions can be generalized to functions with Dunkl-bounded variation. However, we proved the completeness and compactness properties of $BV_k(\Omega)$, the lower semicontinuity, approximation with smooth functions and so on. These results may lay the foundation for the variational theory in the Dunkl setting. We established a version of the Gauss–Green Theorem in the Dunkl case, and we obtained some excellent properties of Dunkl-bounded variation functions from the perspective of capacity. Finally, we developed a heat semigroup characterization of $BV_k$ functions, and consequently obtained an important limit representation relation. We believe that the investigation of these problems will provide a useful tool for the study of potential theory and function spaces.

**Author Contributions:** Writing—original draft preparation, X.M.; Writing—editing, X.M.; methodology, X.M., X.X. and Y.L.; check, X.X.; review, Y.L. All authors have read and agreed to the published version of the manuscript.

**Funding:** Y. Liu was supported by the National Natural Science Foundation of China (No. 11671031).

**Informed Consent Statement:** Not applicable.

**Data Availability Statement:** Not applicable.

**Conflicts of Interest:** The authors declare no conflict of interest.

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
