# Peer review of "Capacity and the Corresponding Heat Semigroup Characterization from Dunkl-Bounded Variation"

_fractalfract, doi:10.3390/fractalfract5040280_

Round 1

Reviewer 1 Report

line 7, rewrite this beginning as: Throughout this paper, $\Omega$ stands for a open subset of $\mathbb R^N$ with $N\ge 2$.

line 35, have studied -> studied

line 138, Theorem 20(iii), it is possible to say something on the equality case of this strong subadditivity; 

between line 185 and line 186, Example 29, three 'if' should be typewritten at the same vertical line, and similar for 'if...' and 'otherwise'

Author Response

Dear reviewer, 

Thank you for your constructive suggestions and comments. We have   revised our manuscript carefully according to your suggestions and comments. Please see the attached for the details.

Best regards,

Yours,

Meng, Xiangling; Liu,Yu & Xie, Xiangyun 

Reviewer 2 Report

The paper is excellent.

Author Response

Dear Reviewer:

We would like to thank you for taking the time and effort to review the manuscript (Manuscript ID: fractalfract-1449242). Thank you very much for your encouraging comment, which has inspired us a lot. According to the suggestions of all reviewers, we have supplemented the abstract and added conclusion. Moreover, we have checked our paper carefully and also polished it. All the changes we have made are highlighted with yellow background in the marked revised manuscript.  

All the best wishes to you.

Sincerely,

Meng, Xiangling; Liu,Yu & Xie, Xiangyun

Reviewer 3 Report

  1. The abstract very short.

  1. -Introduction needs to explain the main contributions of the work more clearly. The difference between present work and previous Works should be highlighted. Comparison with recent study and methods would be appreciated What is the motivation of the proposed work. Also, a detailed literature review has not been written
  2. There is not a conclusion section in article.
  3. The article is not prepared in an easy-to-understand format. For example,

-In the Section 1 of the article, there are no definitions, properties, theorems taken from previous studies. For example, the definition of BV spaces is not available in the study.

-Also, many definitions, theorems, and properties are not cited. For example in section 1, were "a function of bounded variation" and "The weight function wk associated to Dunkl operators"  described for the first time in this article? Or were they taken from another article? If so, why was it not cited. In section 1, is every definition, theorem, and feature that exists in the article first used in this article? Or is the definition, theorems not cited? I did not understand which of these situations true.

Author Response

(The authors gave the same response as above.)

Reviewer 4 Report

In this manuscript, the authors investigated some important basic
properties of Dunkl bounded variation functions and its functional
capacity. They generalized the characterization of a heat semigroup
of the Dunkl BV space, thereby giving its relation to the function of Dunkl bounded variation.\\

The results are interesting. The paper can be considered for
publication in "Fractal and Fractional" after making
some of the following corrections.\\

\begin{enumerate}
\item Some symbols, such as commas and ..., are not used correctly.
 Some of these mistakes are highlighted in the attached file. The authors must correct all of them carefully.\\

\item On page 12, Proposition 16, the second line of Proposition 16,
also in Theorem 17, the third line of Theorem 17, please put "," in
the end of line.\\

\item On page 10, line 2 from below, please put "," in
the end of line, instead of ".".\\

\item On page 12, line 5, the phrase "Therefore," should be replaced by
"therefore,".\\

\item On page 15, line 5, the phrase "then," should be replaced by
"Then,".\\

\item On page 16, line 18, the phrase "Therefore," should be replaced by
"therefore,".\\

\item The same mistakes are abundantly seen in other pages. The
authors
should be carefully correct all of them.\\
\end{enumerate}

Author Response

(The authors gave the same response as above.)

Round 2

Reviewer 3 Report

Deficiencies in the article have been corrected. The article can be accepted in its final form.